# Minor Actinides Transmutation Performance in a Closed Th–U Cycle Based on Molten Chloride Salt Fast Reactor

**DOI:** 10.3390/ma15238555

**Published:** 2022-12-01

**Authors:** Liaoyuan He, Liang Chen, Yong Cui, Shaopeng Xia, Yang Zou

**Affiliations:** 1Shanghai Institute of Applied Physics, Chinese Academy of Sciences, Shanghai 201800, China; 2CAS Innovative Academies in TMSR Energy System, Chinese Academy of Sciences, Shanghai 201800, China; 3University of Chinese Academy of Sciences, Beijing 100049, China

**Keywords:** molten chloride salt fast reactor, Th–U cycle, breeding performance, MA transmutation, online reprocessing

## Abstract

The molten chloride salt fast reactor (MCFR) with a closed Th–U fuel cycle is receiving more and more attention due to its excellent performance, such as high solubility of actinides, superior breeding capacity, and good inherent safety. In this work, the neutronics performances for different minor actinides (MA) loadings and operation modes are analyzed and discussed based on an optimized MCFR. The results indicate that online continuous reprocessing can significantly increase the transmutation performance of MAs. In addition, MA loadings have an obvious effect on the neutronics characteristics of the MCFR, and it is helpful for improving the MA transmutation capability and ^233^U breeding performance, simultaneously. When MA = 5 mol%, the average annual MA transmutation mass and incineration mass can achieve about 53 kg and 13 kg, respectively, and the corresponding annual net production of ^233^U is 250 kg. When MA = 33.5 mol%, the annual MA transmutation mass and incineration mass can be about 310 kg and 77 kg, respectively, and the corresponding annual net production of ^233^U is 349 kg. However, when the MA loadings exceed 10%, the corresponding *k*_eff_ will exceed 1.1 for decades, even if only Th is continuously fed online. The results also indicate that the transmutation ratio (TR) and incineration ratio (IR) of MA increase and reach maximum values in the first decades for all the different MA loadings, which means MA may be fed into the fuel salt to improve its transmutation capability. Moreover, though MA loading will increase the level of radiotoxicity of the core in the early stage of burnup, the radiotoxicity of MA will drop rapidly after a brief rise during the operation. It can also be found that the temperature coefficient of reactivity (TCR) of all different MA loadings can be negative enough to maintain the safety of the MCFR during the whole operation, although it decreases in the beginning of life (BOL) with the increasing MA loading. Furthermore, the evolution of an effective delayed neutron fraction (EDNF) is also researched and discussed, and the EDNF varies most significantly when loading MA = 35.5 mol%, with a range of 273 to 310 pcm over the entire 100 years of operation.

## 1. Introduction

Approximately 20–30 metric tons of spent fuel are produced by a typical 1000 Mwe pressurized water reactor (PWR) per year, and the spent fuel will increase steadily with the release of nuclear energy [1]. Traditional methods for dealing with the spent fuel, such as permanent geological disposal, are very unfavorable for reducing the long-lived radioactivity and improving the utilization of fuel [2]. For solving those problems, the research about transuranic (TRU) transmutation is receiving increased attention. Plutonium, which is contained in mixed-oxide (MOX) fuel, can be reused, while MAs (Np, Am, Cm) that have long-lived radiotoxicity are very difficult to reuse, and they are a major source of high-level radioactive waste [3]. Therefore, research on the effective transmutation of minor actinides (MA) in reactors is of great significance.

MA transmutation has been studied in various kinds of reactors including thermal and fast reactors [4,5,6,7,8,9]. Liu et al. researched the MA transmutation performance with different loading methods in a PWR [4], and the results show that the heterogeneous distributions of MA nuclides can avoid the intense reduction of the initial *k*_eff_ and suggests people may concentrate on the transmutation of ^237^Np in the PWRs to obtain the isotope ^238^Pu required for the production of nuclear batteries and greatly reduce the inventory of long-lived high-level radioactive MAs. In addition, Liu et al. studied MA transmutation performance by coating MAs of varying thicknesses on fuel pellets of the AP1000 [5], which shows that when the MA coating is 0.002 cm and the loading MA mass is about 300.0 kg, the mass of MA nuclides incinerated after a one-year fuel cycle is 44.0 kg, and the corresponding transmutation rate can be 14.7%. Eduardo Martinez et al. discussed MA transmutation capability based on a boiling water reactor (BWR) using a uranium oxide (UO_2_) fuel assembly or a mixed-oxide (MOX) fuel assembly, respectively. It indicates the MA and Pu reduction can be achieved by using each proposed assembly [6].

Besides thermal reactors, research about MA transmutation had also been carried out for fast reactors such as the sodium-cooled fast reactor (SFR) [10], supercritical water-cooled fast reactor (Super FR) [11], and lead-cooled fast reactor (LFR) [12] because of their larger ratio of fission–capture cross-section, a higher neutron flux, and a negative neutron consumption. The research results show that although loading MAs in fast reactors would slightly weaken the fuel’s negative feedback effect, the MA transmutation performance is significantly improved. Therefore, burning MAs in a fast reactor is recognized as a promising method.

The molten salt reactor (MSR), selected as one of the six candidate reactors by the Generation-IV International Forum (GEN-IV), has superior features on inherent safety, neutron economy, and non-proliferation [13]. The flow of liquid fuel allows the MSR to operate with online refueling instead of shutting down the reactor to refuel, and it enables the closed fuel cycle, which will further improve the MA transmutation performance [14,15]. Due to the advantages of MSRs in MA transmutation, a series of studies on MA transmutation capability based on MSRs has been carried out. For instance, O. Ashraf et al. compared the MA transmutation performance in the thermal and fast MSR [16]. It shows that the fast MSR consumes more of the generated Pu isotopes in the fuel salt and has a higher transmutation ratio (TR) in a closed Th–U cycle. Yu et al. researched the MA transmutation capability and neutron performance in the SMSFR for different MA loadings. When MA = 18.17 mol%, the transmutation fraction can achieve about 95% on iso-breeding, and the safety for the SMSFR with all different MA loadings can be guaranteed during the whole operation [3].

Unlike the SD-TMSR and SMSFR, which use fluoride salts as carrier salts, the MCFR, as another type of MSR, uses chloride salt as a carrier salt. Research on the MFSR originated in the 1960s, during which time the Oak Ridge National Laboratory (ORNL) first proposed the Molten Salt Reactor Experiment (MSRE). It had been operated for four years, and the key technologies of fuel flow and material corrosion resistance were verified in the MSRE, which provided some technical reference for the subsequent research of the molten fluoride salt reactor [15]. However, the structural material information underpinning an MCFR is uncertain, corrosion effects form the most difficult problem, and thermodynamic studies suggest the use of molybdenum alloys as structural materials [17]. Various purification methods such as dry chemical treatment, zone refining, and gas sparging have been considered for fluoride and chloride salt systems [18].

However, compared with the molten fluoride salt reactor, the MCFR has a harder neutron spectrum due to the slight slowdown of neutrons in chloride salt [19], and it is helpful for improving the breeding performance and transmutation capability of MAs in an MCFR. In addition, due to the small neutron absorption cross-section under the fast spectrum in an MCFR, the production of fission products (FPs) will be significantly reduced, and the corresponding reprocessing cycle can be appropriately prolonged. Furthermore, more FPs need to be removed in online reprocessing, while in the MCFR some nuclides such as I, Er, and Yb that need to be removed in the molten fluoride salt reactor have a negligible effect on the neutron performance in the MCFR [20]. Fewer nuclides needing to be reprocessed, longer reprocessing period, and less production of FPs all make the reprocessing of the MCFR easier, and this makes a lot of sense for reprocessing technology, which is immature. In addition, the solubility of MAs in the MCFR is much higher than that of the molten fluorine salt reactor; thus, more MAs can be loaded in an MCFR, which will further improve its MA transmutation capability [21]. Thus, the MCFR has great potential for transmuting MAs. 

Research on the MCFR dates back to the 1950s, when the MCFR concept was proposed by the Oak Ridge National Laboratory (ORNL) to research the breeding performance in the U–Pu cycle [19]. In the 1960s, the Atomic Energy Authority in the UK conducted a series of studies on the MCFR [22], which further demonstrated the fuel-breeding ability of the MCFR. Due to advantages in fissile nuclide breeding and MA transmutation, the MCFR is presently receiving increasing attention worldwide. REBUS-3700, which was proposed in France, shows a rare combination of excellent breeding capability and strong inherent safety [23]. In addition, Moltex Energy in the UK and Germany have each proposed a new type of MCFR, and a series of important studies have been carried out based on them [24,25].

In our previous work, a lot of research on the MCFR was carried out, which achieved some results such as ^37^Cl enrichment selection, carrier salt selection, material selection, etc., and thus completed the design of the MCFR [20,26,27]. Moreover, we have developed a code to optimize the MCFR with multiple objectives in equilibrium (EQL) and achieved an optimized MCFR design, based on the optimized model, and the ^233^U breeding capability and other neutronics parameters have been analyzed in different cycle methods [20,28]. Here, we focus ourselves on the MA transmutation capability, ^233^U breeding performance, and safety parameters for different MA loadings and different operation modes based the optimized MCFR to evaluate its potential for transmuting MAs and breeding ^233^U simultaneously and provide rationale for involving MAs in the Th–U cycle. 

## 2. General Description of MCFR and Analysis Methodology

### 2.1. Reprocessing Schemes

Online reprocessing is the main feature and advantage of MSR, and it enables MSR with good neutron economy, high fuel utilization, inherent safety, and makes the closed fuel cycle of MSR possible. There are two main reprocessing systems in MSR: the He bubbling system is used for removing the noble gaseous and metallic FPs continuously online, and the online chemical reprocessing system is applied for reprocessing soluble FPs. The reprocessing diagram is shown in Figure 1 [29], and the detailed steps have been explained in our previous work [28]. With the fluorination and reductive extraction techniques, ^233^U decayed from the ^233^Pa will be collected in a stockpile, the FPs will be removed from the carrier salt, and the carrier salt will be recovered for reuse [30].

### 2.2. Simulation Tool

Unlike the traditional solid-fueled reactors, continuous online refueling and removal of FPs is applied in MSR; thus, the corresponding depletion equation is non-homogeneous and can be described by improving the conventional depletion equation, which can be written as follows: (1)$$\left\{ \begin{array}{l} \frac{dN_{i}(t)}{dt}=\sum_{j}\lambda_{j,i}N_{j}(t)-\lambda_{i}N_{i}(t)+C_{i}\\ \lambda_{j,i}=f_{j\to i}\sigma_{j}\varphi (t)+\gamma_{j\to i}\lambda_{j}^{decay},\;\lambda_{i}=\sigma_{i}\varphi (t)+\lambda_{i}^{decay}+\lambda_{i}^{c} \end{array}\right.$$
where *N_i_*(*t*), *N_j_*(*t*) stand for number density of nuclide *i* and *j*, respectively; *γ_j_*_→*i*_ refers to branching ratio of decay from nuclide *j* into *i*; *f_j_*_→*i*_ indicates the probability of nuclide *j* to *i* by a neutron absorption reaction; *λ_i_*, *λ_j_* represent the decay constant of nuclide *i* and *j*, respectively; $\lambda_{i}^{c}$ is fictive decay constant of nuclide *i* due to the chemical reprocessing; *σ_i_*, *σ_j_* donate absorption cross-sections of nuclide *i* and *j*, respectively; *C_i_* indicates the feed rate of nuclide *i*; $\lambda_{i}^{\mathrm{decay}}$ is the decay constant of nuclide *i*.

For the depletion calculation of MSR, an in-house program called TMCBurnup, which couples the TRITON in the SCALE6.1 program [31] and a novel depletion code MODEC, was developed in our previous work [32,33]. The flowchart of TMCBurnup is shown in Figure 2 [33]. First, the parameters of core geometry and salt are initialized by users. Then, the calculation of neutron transportation and depletion are performed by the TRITON module and MODEC, respectively. Next, ^233^U and ^232^Th fuel need to be refueled into the core, and the total mass and the ratio of reinjected ^232^Th and ^233^U are determined by two restrictive conditions, namely ensuring the total heavy metal inventory constant and keeping the reactor critical. The calculation is performed iteratively until a stop condition set by the user is reached. Due to extremely deep burnup of MSR, traditional depletion codes such as ORIGEN-S may not be accurate in solving this depletion equation in deep burnup [34]. Thus, a novel molten salt reactor specific depletion code (MODEC) was developed in our previous work [32], in which a high-precision depletion solver is embedded to ensure the accurate tracking of the evolution of the nuclides. Two optional methods, augmented matrix and numerical integration, are applied to deal with the non-homogeneous term caused by the continuous feeding characteristic of MSR. The results of different depletion calculation cases showed that MODEC is suitable for the depletion calculation of MSR under different online reprocessing modes [32]. 

To prove the applicability of TMCBurnup in the depletion calculation of a molten salt fast reactor, the evolution of heavy nuclides in MSFR was calculated with different start-up fuels and compared with the reference [35]. The results are shown in Figure 3a,b, and the evolution of heavy nuclides calculated by TMCBurnup is almost identical to the reference results, which prove the reliability of TMCBurnup in the depletion calculation of a molten salt fast reactor. Thus, it can be applied for the depletion calculation of MCFR.

### 2.3. Description of the Optimized MCFR

In our previous work, a lot of basic research about the MCFR was carried out, including the influence of ^37^Cl enrichment on neutronic performance and the selection of a reprocessing method suitable for MCFR, as well as the determination of a carrier salt, referring to the configuration of REBUS-3700 and MSFR [23,36]. Then, the pre-concept design scheme of the MCFR was finished, which is shown in Figure 4. It mainly includes the core and the fertile zone. The fuel salt flows from the bottom to the top in the active core. The blanket consists of 55 mol% and NaCl-45 mol%. ^232^ThCl_4_ surrounds the active core in both the radial and axial direction (blue area in Figure 4) for breeding ^233^U, and these two zones are separated by a Ti-based alloy. Besides the blanket, a graphite reflector (yellow area in Figure 4) is used for fertile fuel saving. Next to the graphite reflector, B_4_C (green area in Figure 4) is applied for absorbing the leaking neutrons. 

In addition, to improve the neutronics performance of the MCFR in EQL, under the fixed power of 2500 MW, the multi-objective optimization of the MCFR was carried out using a script which couples molten salt reactor equilibrium-state analysis code (MESA) and a novel intelligent optimization algorithm, and the corresponding optimized design parameters are listed in Table 1 [20].

## 3. Results and Discussions

To evaluate the MA transmutation performance of the optimized MCFR, the changes of reactivity for different MA loadings at the initial critical are discussed first. Then, the evolution of neutronics parameters during the operation without reprocessing is researched. Finally, the MA transmutation performance, ^233^U breeding capability, and other related neutronics parameters for different MA loadings with online reprocessing and refueling are analyzed.

### 3.1. Reactivity Varying at the Initial Critical

The molar fraction of heavy metal is 45 mol% in the MCFR. When the molar fraction of ^233^U accounts for 5.11 mol% and ^232^Th is 39.89 mol%, the MCFR can be critical. When the mixture of ^232^Th and ^233^U is replaced by a different molar fraction of MA, in the case of keeping the relative ratio of ^233^U and ^232^Th unchanged, the evolution of *k*_eff_ is shown in the Figure 5. 

In Figure 5, MA stands for the MA mixture partitioned from the spent nuclear fuel, the actinide weight ratios of MA are listed in Table 2 [37], and Cm, Am, and Np refer to using the single nuclides of Cm, Am, and Np to replace the mixture fuel, respectively. It is mainly composed of Np, Am, and Cm. It can be found that when the MA mixture, ^237^Np, ^241^Am, or ^243^Am, is used to replace the initial mixture fuel, the *k*_eff_ first decreases and then increases; however, if Cm is used to replace the initial mixture fuel, *k*_eff_ increases monotonically with the increase of MA loading.

The evolution of the contribution of main nuclides to *k*_eff_ with the increase of the MA loading is shown in Figure 6, where Am refers to ^241^Am, Np indicates ^237^Np, and Cm donates ^245^Cm. It can be noticed that since the fission cross-section of ^232^Th is slight, it hardly affects the *k*_eff_. Therefore, *k*_eff_ is mainly determined by the fission reaction rate of ^233^U and MA. It can be also found that the fission reaction rate of ^233^U declines with the increasing of MA loadings due to the decreased ^233^U mass and the decreased mean microscopic fission cross-section of ^233^U caused by spectrum hardening (see Figure 7). The increase in the fission reaction rate of ^245^Cm is always greater than that of the decrease in fission rate of ^233^U with the increase of ^245^Cm loading due to its large fission cross-section and excellent fission capability (see Figure 7). However, since the fission cross-section of ^237^Np and ^241^Am is significantly smaller than ^233^U, the *k*_eff_ declines first with the increasing of ^237^Np (^241^Am) loading to a minimum value at about ^237^Np (^241^Am) = 10 mol%. Afterwards, as the mass of MA increases and neutron spectrum hardens, the increase of fission reaction rate from MA is greater than the decrease of the fission reaction rate from ^233^U, which leads to the gradual increase of *k*_eff_. 

### 3.2. MA Transmutation without Reprocessing and Refueling

In this subsection, the MA transmutation performance in different MA loadings is researched without reprocessing and refueling. When the MA = 0 mol%, the initial loading’s ^233^U accounts for 7.00% to ensure enough excess reactivity. When using different molar fractions of MA to replace the mixture of ^232^Th and ^233^U, in the case of keeping the relative ratio of ^233^U and ^232^Th unchanged, the evolution of *k*_eff_ is shown in Figure 8. It can be found that the varieties of *k*_eff_ with different MA loadings are basically the same as those described in Section 3.1; the *k*_eff_ first decreases and then gradually increases with the increase of MA loading. While the lifetime of the MCFR increases with the increasing of MA loading, though before MA < 10 mol%, the initial *k*_eff_ decreases when the MA loading increases. In addition, the more MA loading in the initial there is, the slower the *k*_eff_ declines during the operation. When the initial loading of MA is greater than 10%, the corresponding *k*_eff_ even increases in the early stage with burnup. This is due to the fact that the MA is mainly composed of ^237^Np and ^241^Am, and a considerable amount of Pu can be produced from the following approaches:^237^Np + n → ^238^Np → *β*^−^ + ^238^Pu
Am241+n→242Am→{β−+Cm242→α+Pu238β++Pu242

Thus, as MA loading increases, the reactivity loss decreases or even increases with burnup due to the prompt reactivity provided by the MA itself and the delayed reactivity released by the produced Pu isotopes.

In general, the transmutation performance of MA can be judged by the transmutation ratio (TR), which is defined as Equation (2), and it stands for the ratio of MA disappeared to MA loading.
(2)$$TR(t)=1-\frac{M{\left(t\right)}_{}}{M(B)+M_{feed}(t)}$$

Here, *M*(*t*) refers to the MA inventory at operating time *t*, and *M*(*B*) and *M_feed_*(*t*) stand for the total mass of loaded MA and MA fed online at time *t*. In this work, only ^232^Th and ^233^U are fed online; thus, the *TR* could be simplified as: (3)$$TR(t)=1-\frac{M{\left(t\right)}_{}}{M(B)}$$

As we all know, MA transmutation can be realized with either neutron capture or neutron fission; however, MA can only be completely converted into short-lived nuclides by neutron fission, so as to effectively eliminate its long-lived radioactive hazards. Thus, some views suggest that only the fission reaction is the effective way for MA transmutation. In order to truly reflect the relationship between the fissioned MA and the total loading of MA, the incineration ratio (IR) of MA is introduced, and since no MA is fed online during the whole operation, IR can be expressed in Equation (4):(4)$$IR(t)=1-\frac{\sum_{i}\left(M_{i}(B)-M_{i}(t\right))*\frac{R_{f(i)}(t)}{R_{a(i)}(t)}}{\sum_{i}M_{i}(B)}$$
where the *R*_*f*(*i*)_(*t*) and *R*_*a*(*i*)_(*t*) refer to the neutron fission rate and the neutron absorption rate for nuclide *i* at operating time *t*.

The evolution of TR and IR of MA with different MA loadings is shown in Figure 9a,b, and more details about neutronics performance after 100 years of burnup are listed in Table 3. It can be noticed that TR and IR increase gradually with the depletion. With the same burnup time, the smaller the initial loading of MA is, the larger its corresponding TR and IR are. In addition, as the MA loading increases, the TR and IR in EQL also increase, which is mainly due to increased MA loading leading to a longer lifetime of operation. Furthermore, except ^245^Cm, other MA can all be effectively transmuted in the MCFR without reprocessing and refueling. This is due to the fact that ^245^Cm is at the bottom end of the depletion chains of actinides, and its accumulation from Am and Pu isotopes is greater than the consumption during the operation. In general, the incineration efficiency of MA in MCFR is obviously higher than that in thermal reactors due to the higher neutron flux and significantly larger ratio of a fission–capture cross [5,6,7].

### 3.3. MA Transmutation with Online Reprocessing and Refueling

In MSR, online refueling and reprocessing is a unique advantage that can improve the utilization of fissile nuclides, simplify the reactor control requirements, and achieve a closed fuel cycle [2]. Thus, it is of great significance to analyze the MA transmutation under the online reprocessing and refueling condition. In this subsection, the MA transmutation capability and other important neutronics performances in the MCFR with reprocessing and refueling are researched. The online reprocessing and feeding methods presented in subsection II.A and II.B were applied.

In the calculation, a 238-group ENDF-B/VII.0 cross-section library was used, there were 10,000 neutrons per *k*_eff_ cycle, the first 50 *k*_eff_ cycles was skipped, and a total of 250 *k*_eff_ cycles were run in a criticality calculation with an average error of about 40 pcm. 

The required ^233^U mass at the critical condition (*k*_eff_ = 1) for different MA loadings is listed in Table 4. To keep the critical condition (*k*_eff_ = 1), the maximum initial MA loading for the MCFR is calculated to be 33.5 mol%; hence, the corresponding ^233^U loading is 0 mol%.

#### 3.3.1. Evolution of *k*_eff_ and EALF

During the whole operation, the ^233^U and ^232^Th are fed online, the critical condition set in this work is *k*_eff_ = 1.00–1.02, and a total of 10 cases including the loading MA = 0%, 1.0%, 3.0%, 4%, 5%, 7%, 10.0%, 20.0%, 30.0%, and 33.5% were calculated. The evolution of *k*_eff_ during the whole operation is shown in Figure 10. In order to clearly show the results under different MA loadings, only the most typical five cases are shown. 

It can be found that when loading MA = 10 mol%, the corresponding maximum *k*_eff_ value is just about 1.02 during the whole operation. If the initial MA loading is more than 10%, the *k*_eff_ exceeds the upper limit of 1.02 in about 10 years due to the gradual release of reactivity from the accumulated Pu during the burnup, even if only Th is fed online during the burnup, which means that when the MA loading exceeds 10 mol%, only relying on the method of continuous online refueling and reprocessing cannot ensure that the core is always critical, and it is necessary to sacrifice some neutron economy to make the core critical by using control rods and burnable poisons.

Neutron spectra which can help fundamentally indicate the evolution of some important neutronic parameters such as TCR, breeding ratio (BR), and TR have great research significance. In order to give a quantitative description for the neutron spectra to research the evolution of neutron spectra during the operation, the energy of the average lethargy-causing fission (EALF) is introduced [38], which can be defined as:(5)$$EALF=\exp\left\{\frac{\int \ln E\phi (E)\sum_{f}(E)dE}{\int \phi (E)\sum_{f}(E)dE}\right\}$$
where *ϕ*(*E*) and ∑*_f_*(*E*) refer to the energy-dependent neutron flux and macroscopic fission cross-section, respectively. The EALF during the whole operation is shown in Figure 10, and we can notice that when the loading of MA increases, its corresponding initial neutron spectra hardens, which is due to the fact that when the MA loading increases, the number of neutrons in the low-energy region decreases, and the fission cross-section of MA nuclides in the high-energy region will increase rapidly. With the burnup, due to the accumulation of FPs and the transmutation of MA, the EALF gradually decreases, and with the further deepening of burnup, the nuclides of the core for different MA loadings gradually tend to be consistent, and so does the EALF. 

#### 3.3.2. Evolution of Heavy Nuclides and Radiotoxicity

The depletion calculation for different MA loadings is calculated for 100 years. For a clear comparison, only the evolutions of heavy nuclides for MA = 0 mol% (solid lines) and MA = 30.0 mol% (dash lines) are shown in Figure 11. It can be found that the elements of Th, Pa, and U reach the EQL faster than the others: they reach EQL almost in the first 40 years. Furthermore, it should also be pointed out that for the MA = 0 mol% case, only small amounts of MA and Pu are produced in the Th–U cycle during the whole 100-year operation, and the inventories of Th, U, and Pa are stable because of the online refueling and extracting ^233^Pa. As for the MA = 30.0% case, the mass of the MA decrease monotonically results from the neutron fission and capture reactions. The mass of Pu increases rapidly at the beginning due to the neutron captures of MA and then declines caused by their fission consumption. The evolution tendency of Pa is almost the same as that of Th, and the mass of them increases gradually until EQL is reached, but the amount of ^233^Pa is much smaller than Th, since it is the direct neutron capture production of the ^232^Th. Different from the evolution of other nuclides, the mass of U declines during the first 10 years due to the fission depletions and then increases to EQL, which results from the fission of Pu and the capture reaction of ^232^Th.

Mas are the main source of long-lived radioactivity. Thus, the radiotoxicity and decay heat of the MA are calculated and discussed to provide relevant parameters for the intermediate and final storage of nuclear waste. The radiotoxicity and decay heat can be expressed in Equations (6) and (7):(6)$$R(t)=\sum_{i}R_{i}(t)=\sum_{i}r_{i}\lambda_{i}N_{i}(t)$$
(7)$$D(t)=\sum_{i}^{n}D_{i}(t)=\sum \lambda_{i}N_{i}(t)(E_{\beta i}+E_{\lambda i})$$where *N*(*t*) represents the number of nuclei at time *t*, λ_i_ refers to the decay constant, *E_β_* and *E_γ_* are the radiation energy of beta decay and gamma decay, and *r*_i_ indicates the conversion factor of radioactivity to radiotoxicity. From the plot in Figure 12, it can be noticed that since Mas are not initially loaded in the core, both the decay heat and radiotoxicity increase gradually with the burnup for the MA = 0 mol% case, whereas in other cases the corresponding radiotoxicity and decay heat increase rapidly at the beginning and then decrease with the deepening of burnup.

For comparison, the separate contribution of Np, Am, ad Cm to radiotoxicity and decay heat for the MA = 0 mol% and MA = 30.0 mol% cases are shown in Figure 13a,b, and it can be found that for the MA = 0 mol% case that the radiotoxicity and decay heat increase during the burnup, both the radiotoxicity and decay heat are mostly derived from the accumulated Np, and ^237^Np contributes 99.23% to the decay heat and 99.57% to radiotoxicity. As for the MA = 30.0 mol% case, Cm acts as the main resource of radiotoxicity and decay heat, especially ^242^Cm and ^244^Cm, which contribute 39.56% and 57.47% to decay heat, respectively, and they are responsible for 5.99% and 92.93% of the radiotoxicity. Due to the accumulation of ^242^Cm and ^244^Cm caused by the (*n*,γ) reaction of ^241^Am and ^243^Am, the radiotoxicity and decay heat increase significantly in the initial stage and then decrease rapidly with the decay and the transmutation of ^242^Cm (T_1/2_ = 162 days), ^244^Cm (T_1/2_ = 18.1 years), and other Mas. 

#### 3.3.3. ^233^U Breeding Capability

In general, the breeding performance of MSR is evaluated by BR, net production of ^233^U, and doubling time (DT). The BR indicates the ratio of the generation rate of fissile nuclides to the consumption rate of fissile nuclides, and it can be expressed in Equation (8):(8)BR=Rc(U234+Th232+Pu240+U238−Pa233)Ra(U235+Pu241+U233+Pu239)
where *R_c_* and *R_a_* indicate the neutron capture rate and absorption rate. Since this work focuses on the breeding performance of ^233^U, for reflecting the relationship between the production and consumption rate of ^233^U directly, the regeneration ratio (RR) is defined, which is expressed as:(9)RR=Rc(Th232−Pa233)Ra(U233)

Due to the MCFR operating in the continuous online reprocessing and refueling mode, the fed ^233^U should also be included in the calculation of the net production of ^233^U. Thus, the actual net production of ^233^U can be expressed as:(10)M(t)=MinU233(t)+MexU233(t)+MexP233a(t)−MreU233(t)−MBolU233
where ex and re stand for extracting and refueling, respectively, *in* refers to the remaining nuclides in the core and blanket, and *Bol* indicates initial loading at the beginning of lifetime.

The evolution of RR and net production of ^233^U are shown in Figure 14, and the corresponding relative fission rate of the single fissile nuclides are shown in Figure 15. It can be found that the RR decreases gradually for MA = 33.5 mol%, which is mainly due to the fact that fission reaction almost originates from MA and Pu initially, and then ^233^U gradually becomes the main fissile nuclide with depletion, causing the fission consumption of ^233^U to increase, which leads to the increase of RR. Regarding MA = 5 mol%, 10.0 mol%, and 20.0 mol%, the RR increase in about the first 15 years due to the increased fission reaction rate of Pu, with the fission consumption of MA and Pu, ^233^U becomes the main fissile nuclide, and its fission consumption increases gradually, so the RR gradually decreases. For MA = 0 mol%, the MA and Pu produced during the burnup reduce the fission reaction rate of ^233^U, while the production of MA and Pu is relatively low; thus, the increase in RR is not very obvious. The average annual net production of ^233^U is 214 kg and 349 kg for MA = 0% and 33.5%, and the DT is 27.8 years and 7.9 years for the MA = 0% and MA = 30% cases (the initial ^233^U loading for MA = 33.5% is 0 kg, and there is no doubling time for it). 

Figure 16a,b show the separate contributions to the ^233^U production for MA = 0 mol% and MA = 20 mol% for comparison, respectively. It can be found that the net production of ^233^U mainly come from the blanket, and it is almost the same for MA = 0 mol% and MA = 20 mol%. The main reason why the net production of ^233^U corresponding to MA = 20 mol% is higher than that of MA = 0 mol% is that the mass of feeding ^233^U is significantly smaller in MA = 20 mol%, which results from the fact that more MA and the produced Pu act as the main fission nuclides in MA = 20 mol%, reducing the consumption of ^233^U, and it leads to a significant increase of the ^233^U net production.

#### 3.3.4. MA Transmutation Capability

In this subsection, the transmutation performance of MA for the different MA loadings is analyzed, and the plot in Figure 17a shows the evolution of the transmuted MA (solid lines) and incinerated MA (dash lines), while the TR (solid lines) and IR (dash lines) for different MA loadings are shown in Figure 17b. We can find that the total mass of transmuted and incinerated MA, TR, and IR all gradually increase until achieving EQL. Furthermore, increasing the initial loading of MA is beneficial to improving the transmutation and incineration mass of MA in EQL, while the TR and the IR for different MA loadings are almost the same, and the values of TR and IR for different MA loadings are all close to 0.99 and 0.23 in EQL. In addition, in the whole operation, the transmuted and incinerated MA for all different MA loadings have reached the maximum. The smaller the initial loading of MA, the earlier the MA transmutation reaches the maximum. For the case of MA = 5 mol%, the transmuted MA reaches the maximum value around 26 years, while it needs about 72 years for the transmuted MA to reach the maximum value when MA = 33.5 mol%. When we divide the transmuted and incinerated MA by their respective time needed for reaching EQL, the corresponding effective annual average transmutation (incineration) mass of MA is 167 kg/a (47 kg/a), 250 kg/a (67 kg/a), 364 kg/a (90 kg/a), and 516 kg/a (123 kg/a) for MA loadings ranging from 5 mol%–33.5 mol%, respectively. Furthermore, according to the above research, it can be found that if MA can be fueled online after the transmutation of MA reaches the maximum, the transmutation and incineration of MA will be further improved. Therefore, relevant studies were carried out when MA loadings were 5%. The results showed that, if MA is fueled online after reaching EQL, the transmutation of MA can reach 16,850 kg during the 100 years of operation, which is almost three times as high as when fueling ^233^U and ^232^Th online during the whole operation; nevertheless, the corresponding *k*_eff_ could not be maintained between 1.00 and 1.02 only by online fueling and reprocessing, and it even exceeded 1.20 for a time during the burnup.

Then, the transmutation capability of each nuclide of MA is studied separately, and the plots in Figure 18a,b show the evolution of transmutation parameters related to Np, Am, and Cm for the case of MA = 10 mol%. It can be noticed that except for ^245^Cm, the total transmuted (incinerated) mass and TR (IR) of other MAs increase evenly with burnup until reaching EQL, and the transmuted and incinerated ^245^Cm are negative during the first 50 years. Then, with the deepening of burnup, ^245^Cm begins to be truly transmuted and eventually achieves positive IR and TR. In addition, the final TR of all the MAs is nearly the same, while due to the difference in the fission-to-capture ratio of each MA, the IR is different for different MA. The IR of ^243^Cm is the highest, and it is the lowest of ^241^Am and ^243^Am. Although the transmuted mass of ^244^Cm is smaller than that of ^243^Am, its corresponding total incinerated mass is indeed higher than that of ^243^Am.

#### 3.3.5. Safety Parameters

In this subsection, two important parameters, temperature coefficient of reactivity (TCR) and ENDF (*β_eff_*), are researched from the beginning of life (BOL) to EQL. During the whole operation, TCR must be negative enough to ensure the safety of the reactor. Furthermore, the *β_eff_* acts as an important role in reactor operation control; *β_eff_* being too small or fluctuations of *β_eff_* too severe during the whole operation are detrimental to the safety of the core. For the MCFR, the TCR is only affected by the changes of fuel salt and the expansion of core structures due to the fact that there is no moderator in the MCFR. In addition, the change of temperature of the fuel salt is obvious and quick, while a much smaller and slower temperature change happens to the core structure, and the expansion coefficient of the structures is much lower than fuel salt; thus, the effect of structure expansion can be ignored [39]. So, the TCR in the MCFR can be defined as:(11)$$\frac{dK}{dT}total=\frac{dK}{dT}(fuel\;density+\frac{dK}{dT})fuel\;doppler$$
where *K* and *T* refer to the *k*_eff_ and the temperature, respectively. The evolution of TCR for different MA loadings is shown in Figure 19, and it can be found that the constant value of the TCR for MA = 33.5% is the smallest at BOL, and the TCRs of different MA loadings are almost the same in EQL. Due to the fact that the expansion coefficient and density of fuel salt for different MA loadings are the same, the Doppler effect, which is driven by resonance capture in the fertile isotope, is predominant, and the increasing of MA hardens the neutron spectra. MA has a flatter fission cross-section as compared to ^233^U in the neutron spectra of the MCFR; thus, hardening of spectra has a less negative effect on MA fission rather than on ^233^U. In addition, MAs also have a steeper capture cross-section compared to ^233^U, which suggests a higher decrement of captures with the hardening of neutron spectra, thus positively affecting the TCR. So, an increase in the initial loading of MA decreases the absolute value of TCR. In addition, we can find that with the deepening of burnup, the absolute value of TCR corresponding to the low initial MA loadings first gradually increases, while the absolute value of TCR corresponding to the high initial MA loadings decreases in the first years. This is caused by the fact that MA is gradually transmuted with burnup, and ^233^U gradually becomes the main fissile nuclide, which has a negative effect on TCR. In addition, the FPs which have a decreased cross-section with spectra hardening will be accumulated during the burnup; hence, the higher FPs content suggests a higher decrement of captures following spectral hardening, which leads to increased reactivity, thus positively affecting the TCR. Under the combined effect of these two effects, the TCR for low MA loadings gradually increases to EQL, while the TCR for high MA loadings decreases in the early stage of burnup and then gradually oscillates slightly in EQL. In general, the TCR corresponding to different MA loadings can maintain sufficient negativity during the whole operation, which ensures the safety of the MCFR.

*β_eff_* is mainly determined by the fission rate fraction and the delayed neutron fraction of the single fissionable isotopes. It can be expressed as:(12)$$\beta_{eff}=\frac{\sum_{i}\bar{\nu_{d}}\left(i\right)\cdot R_{f}\left(i\right)}{\sum_{i}\left(\bar{\nu_{d}}\left(i\right)+\bar{\nu_{p}}\left(i\right)\right)\cdot R_{f}\left(i\right)}$$
where *v_d_*(*i*) and *v_p_*(*i*) refer to the average delayed neutrons and the prompt neutrons per fission for nuclide *i*, respectively, and *R_f_*(*i*) stands for the fission rate of nuclide *i*. Primary fission nuclides including ^233^U, ^234^U, and MAs are taken into account to ensure the accuracy of *β_eff_*. Furthermore, in order to clearly study the evolution of *β_eff_*, the actual contribution of single nuclides *i* to *β_eff_* is introduced, and it can be defined as:(13)$$\beta_{s}\left(i\right)=\frac{\bar{\nu_{d}}\left(i\right)\cdot R_{f}\left(i\right)}{\sum_{i}\left(\bar{\nu_{d}}\left(i\right)+\bar{\nu_{p}}\left(i\right)\right)\cdot R_{f}\left(i\right)}$$

The plot in Figure 20 shows the evolution of *β_eff_* for different MA loadings, and it can be found that the greater the initial loading of MA, the smaller the corresponding value of *β_eff_* at BOL, and the *β_eff_* for different MA loadings declines to a minimum value first and then rebounds, except for MA = 0 mol%. To explore the source of the variation of *β_eff_*, the *β_eff_* for a single nuclide (*β_I_*) and actual contribution of single nuclides to *β_eff_* (*β_s_*) were calculated, and the details are listed in Table 5 and shown in Figure 21.

It can be noticed that the inherent *β_eff_* (*β_I_*) of nuclides decreases a little due to the hardening of neutron spectra caused by the increases of MA loadings. In addition, except for ^237^Np, the *β_I_* of other MAs is significantly lower than that of ^233^U, and those MAs contribute less to the total *β_eff_* (see Figure 21). With the increasing of MA loadings, the contribution of ^233^U to *β_eff_* decreases due to the decrease of the mean fission cross-section from the spectrum hardening and decreased loading. However, the contribution of MA increases gradually with the increasing of the MA loading, because the increment of the MA (*β_s_*) is smaller than the decrement of the ^233^U (*β_s_*); thus, the *β_eff_* decreases monotonically with the increasing MA loading at BOL. 

The evolution of the *β_s_* of main nuclides for MA = 0 mol% and MA = 5 mol% is calculated and shown in Figure 21, and it can be found that ^233^U and ^237^Np offer the greatest and the second greatest contributions to the total *β_eff_* at BOL, respectively. For MA = 0 mol%, the contribution of ^233^U to *β_eff_* decreases slightly with the depletion, but the contribution of the ^235^U increases the *β_I_* of ^235^U up to about 668 pcm; therefore, the total *β_eff_* rises slightly to EQL. For MA = 5 mol%, during the first 12 years, the total *β_eff_* decreases due to the consumption of ^237^Np. Then, with the online refueling, ^233^U gradually becomes the dominating fission nuclide; thus, the corresponding *β_eff_* gradually increases to EQL.

The changes of *β_eff_* for different MA loadings during the whole operation are listed in Table 6, and with the increasing of MA loading, its corresponding minimum value of *β_eff_* decreases, and the fluctuation of *β_eff_* increases. When MA = 33.5 mol%, the fluctuation value of *β_eff_* increases to 37 pcm during the whole operation, while the maximum value of *β_eff_* is almost the same and equal to its value in EQL due to the fact that the nuclides in EQL are almost same for different MA loadings. 

## 4. Conclusions

The neutronics characteristics, especially the ^233^U breeding performance, the MA transmutation capability, and safety of a 2500 MWth optimized MCFR, are analyzed and evaluated for different MA loadings and different cycle modes, and this research demonstrates the advantages and potential problems of transmuting MAs and breeding ^233^U in the MCFR. It also provides a potentially superior way to achieve Th–U cycle. 

It has been found that if the ratio of ^233^U to ^232^Th is kept unchanged, the initial *k*_eff_ decreases with the increasing MA loading to a minimum value at about MA = 10 mol% and then increases. In addition, in the no-refueling mode, with the increasing MA loading, the burnup will be deepened, and it subsequently improves the MA transmutation performance; all other MAs can be transmuted effectively in this mode except for ^245^Cm.

Furthermore, MA loading has an obvious effect on the neutronics characteristics of the MCFR with online refueling of ^233^U and ^232^Th, and initially loading MA is helpful for improving the MA transmutation capability and ^233^U breeding performance, simultaneously. In addition, online feeding and refueling can significantly improve the transmutation performance of MAs compared to the no-refueling mode. However, when the MA loading is more than 10 mol%, the reactor could not maintain criticality by online fueling and reprocessing alone, and it will exceed the preset *k*_eff_ upper limit of 1.01 in the first years and continue for several decades, even if only ^232^Th is fed online. When MA = 33.5 mol%, the net production of ^233^U can reach about 34,900 kg, and the net mass of the transmuted MAs is about 31,000 kg during the whole operation. There is still a potential approach for further improving the transmutation capability and saving ^233^U by considering feeding MAs to the fuel salt online. However, it is necessary to be aware of the drastic increase in reactivity caused by the feeding of MAs. Both TCR and *β_eff_* decrease with the initial increasing of MA loading. During the operation, the maximum value of *β_eff_* fluctuation is 37 pcm when MA = 33.5 mol%, and the TCR for different MA loadings trends to be a constant of about −7 pcm/K, which is negative enough for the safety.

On a deeper level, the research shows that in addition to using ^233^U, ^235^U, and ^239^Pu as starting fuel, the mixture of MAs and ^232^Th can also be used as the starting fuel to realize Th–U fuel cycle. Using MAs as the starting fuel can not only improve the breeding performance obviously but also reduce the high-level radioactive waste. Therefore, it is strongly recommended to load an appropriate number of MAs in a closed fuel cycle, under the premise of ensuring safety, to improve the breeding performance of the reactor and transmute more MAs.

## Figures and Tables

**Figure 1 materials-15-08555-f001:**
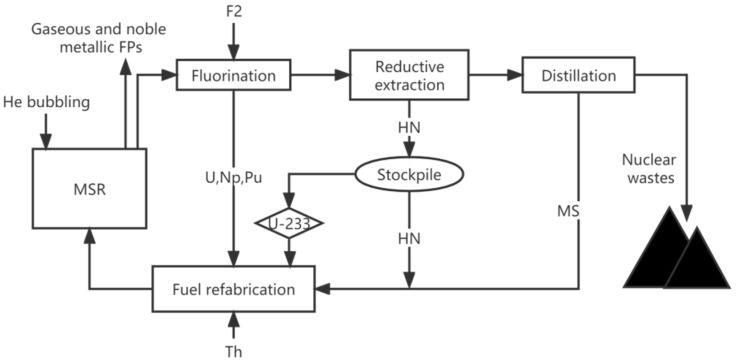
The reprocessing diagram of MSR.

**Figure 2 materials-15-08555-f002:**
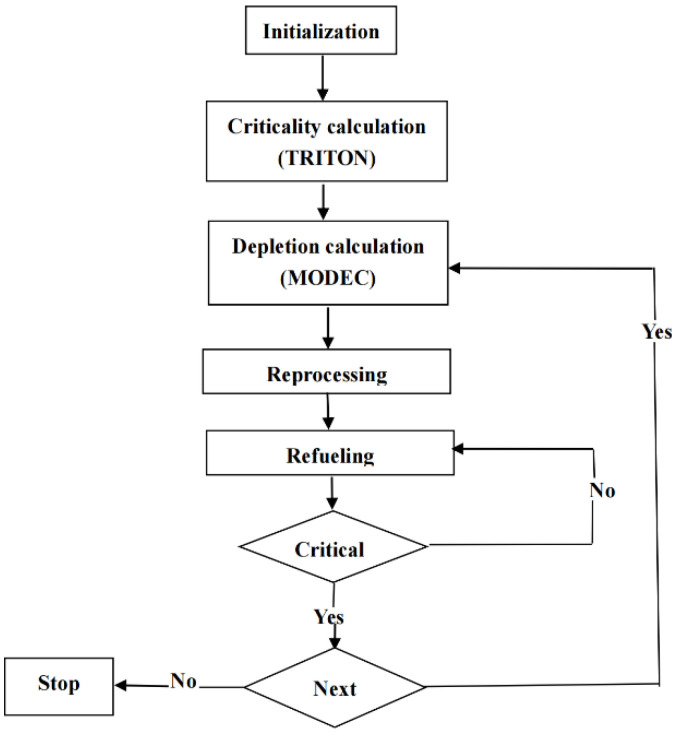
Flowchart of the couple program TMCBurnup.

**Figure 3 materials-15-08555-f003:**
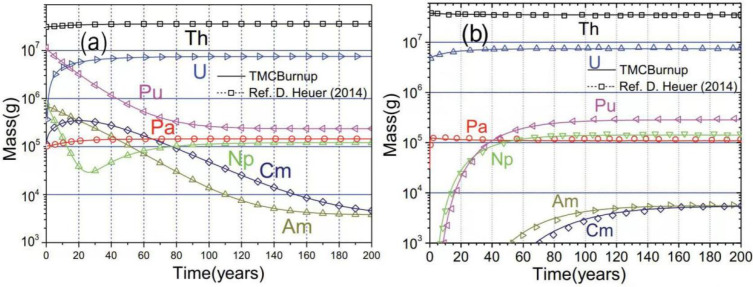
Evolution of heavy nuclides for the MSFR started with TRU (**a**) and ^233^U (**b**) [28,35].

**Figure 4 materials-15-08555-f004:**
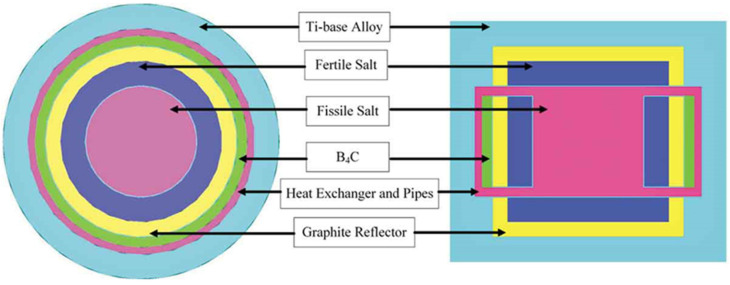
Cross and vertical sections of MCFR.

**Figure 5 materials-15-08555-f005:**
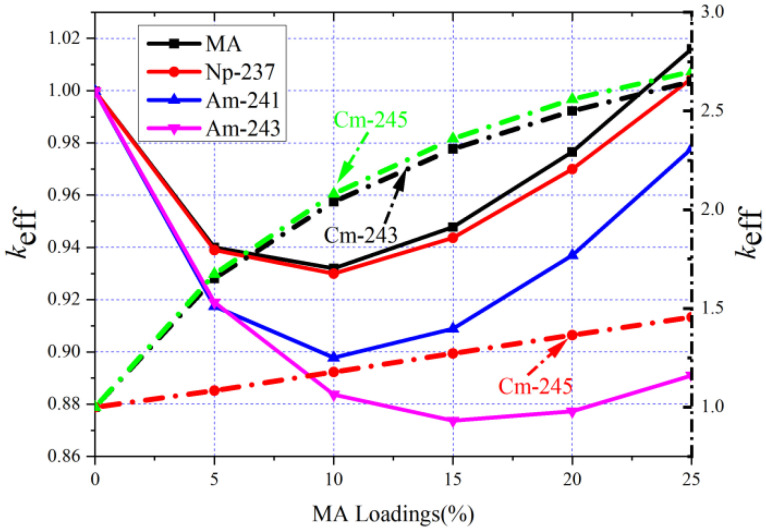
Initial *k*_eff_ for different MA loadings (solid lines indicate Np-237, Am-241, Am-243, and MA mixture and dotted lines stand for Cm).

**Figure 6 materials-15-08555-f006:**
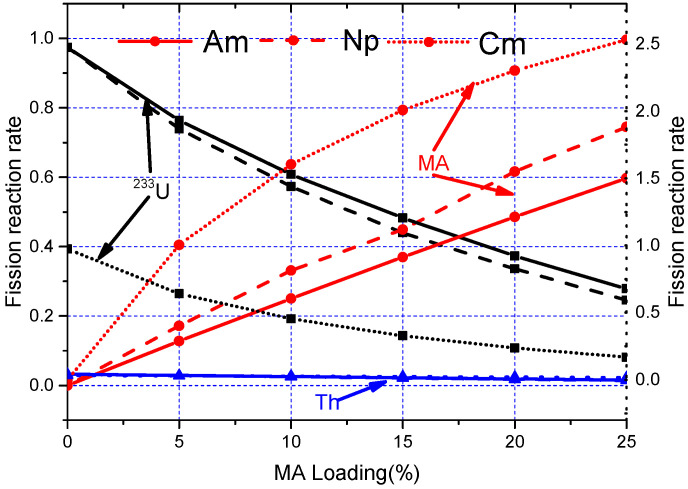
Evolution of fission reaction rate.

**Figure 7 materials-15-08555-f007:**
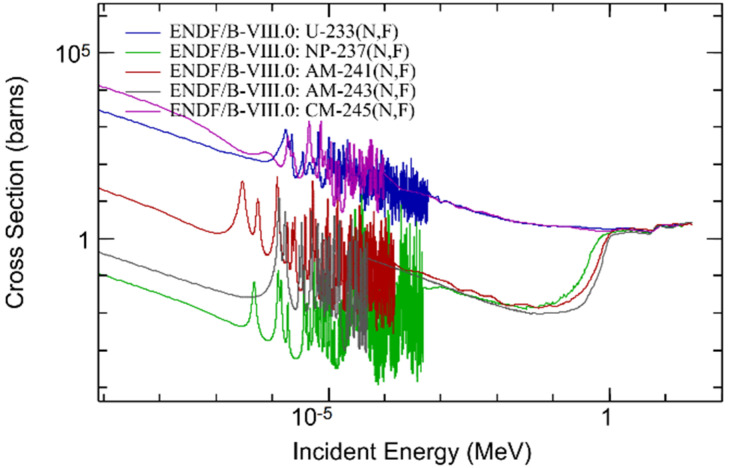
Fission cross-section of nuclides.

**Figure 8 materials-15-08555-f008:**
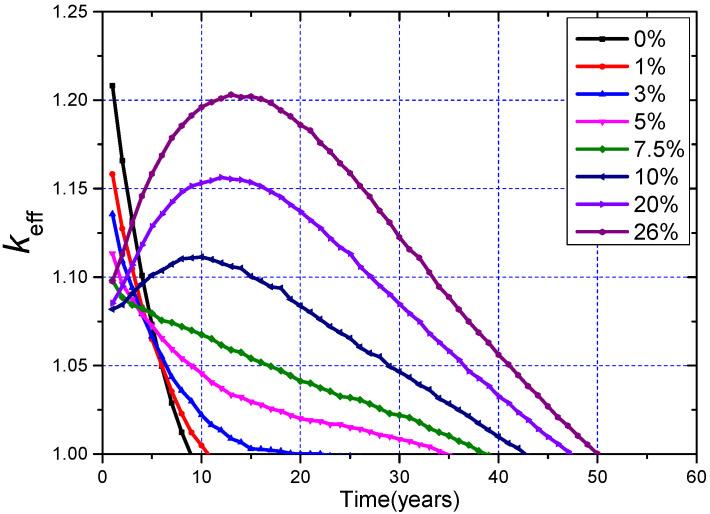
Evolution of *k*_eff_ versus time.

**Figure 9 materials-15-08555-f009:**
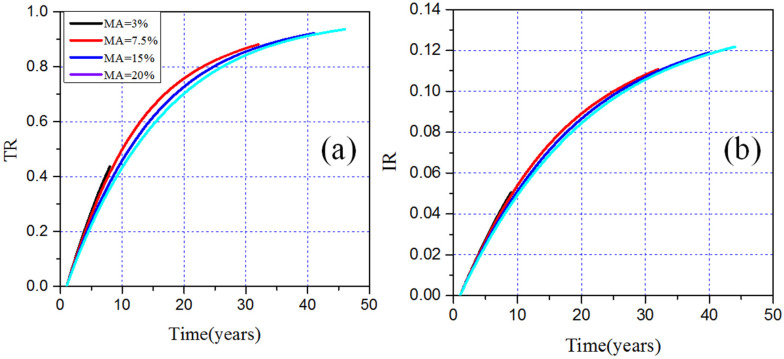
TR (**a**) and IR (**b**) for different MA loadings versus time.

**Figure 10 materials-15-08555-f010:**
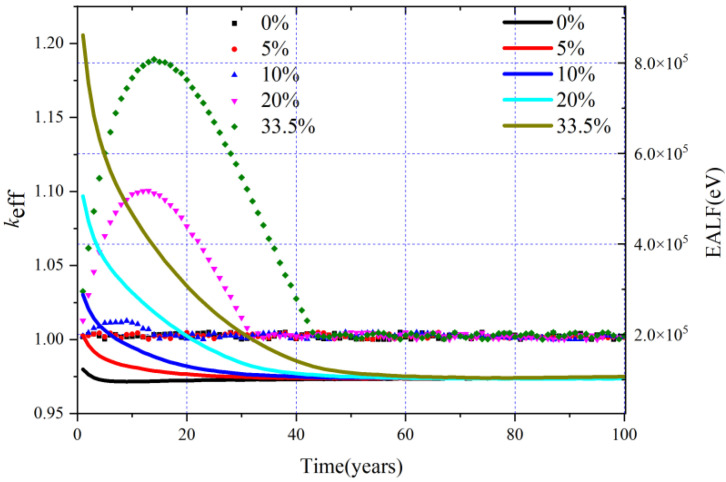
Evolution of *k*_eff_ and EALF for different MA loadings.

**Figure 11 materials-15-08555-f011:**
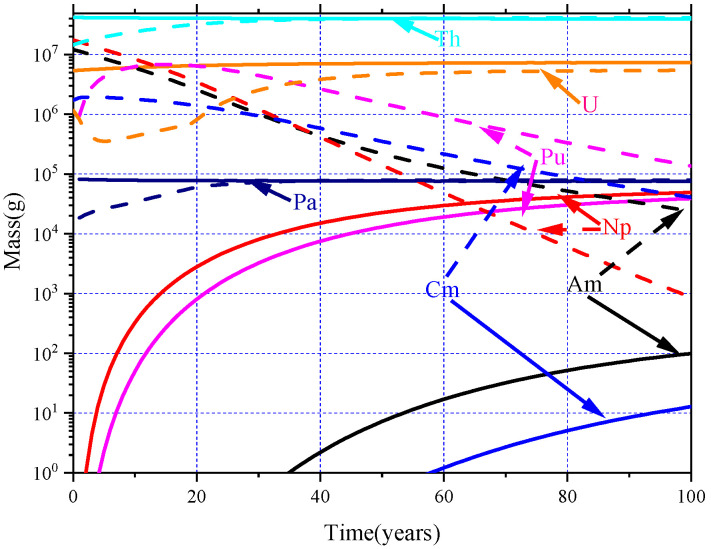
Evolution of heavy nuclides in the core.

**Figure 12 materials-15-08555-f012:**
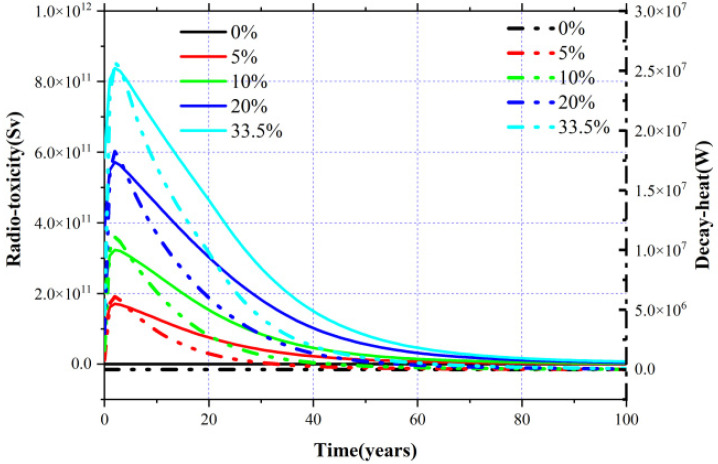
Evolution of radiotoxicity and decay heat for different MA loadings.

**Figure 13 materials-15-08555-f013:**
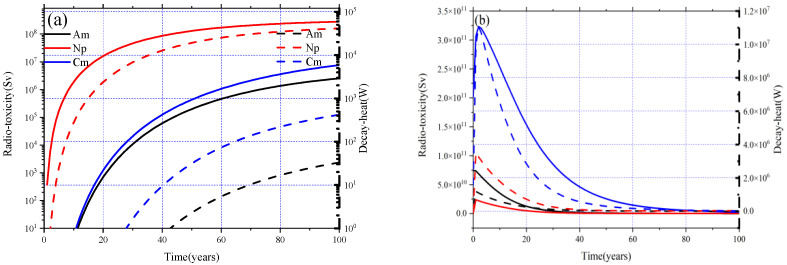
Separate contribution of MA to radiotoxicity and decay heat for MA = 0% (**a**) and MA = 30% (**b**).

**Figure 14 materials-15-08555-f014:**
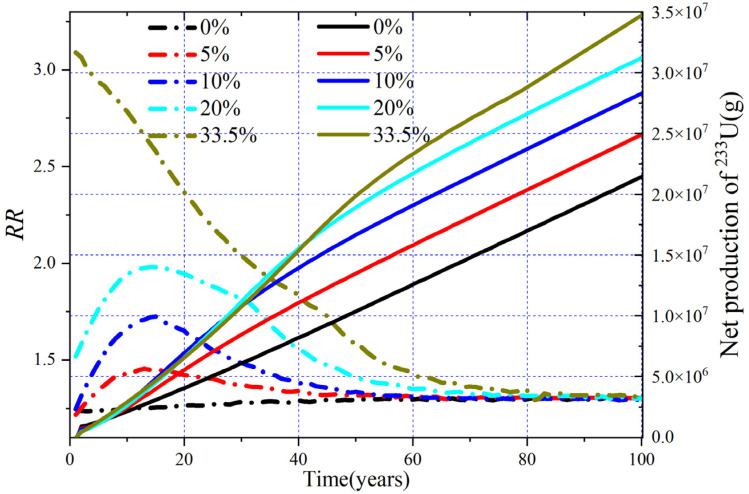
Evolution of RR and produced ^233^U.

**Figure 15 materials-15-08555-f015:**
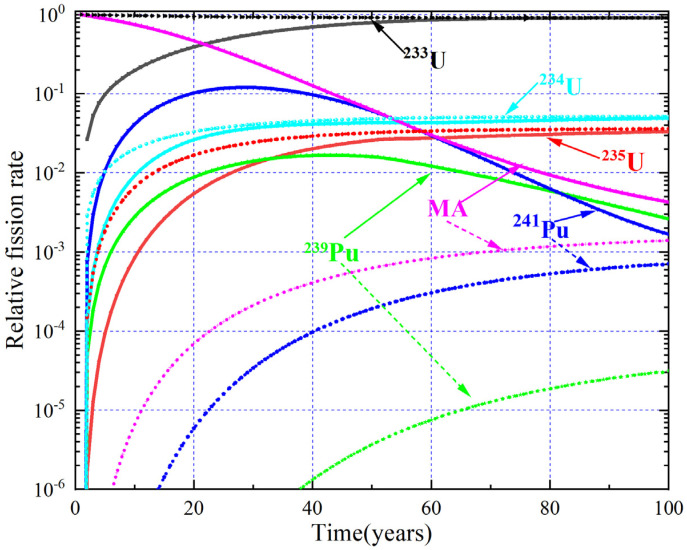
Relative fission rate versus time.

**Figure 16 materials-15-08555-f016:**
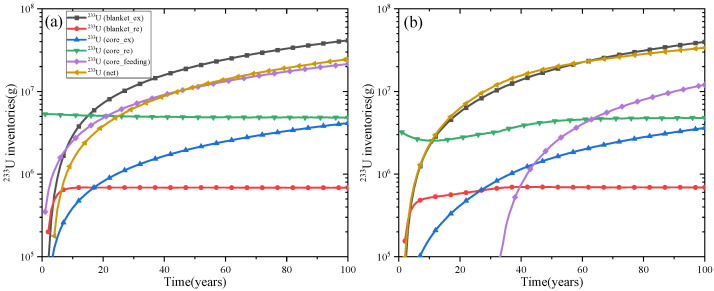
Separate contributions to the ^233^U production for MA = 0% (**a**) and MA = 20% (**b**).

**Figure 17 materials-15-08555-f017:**
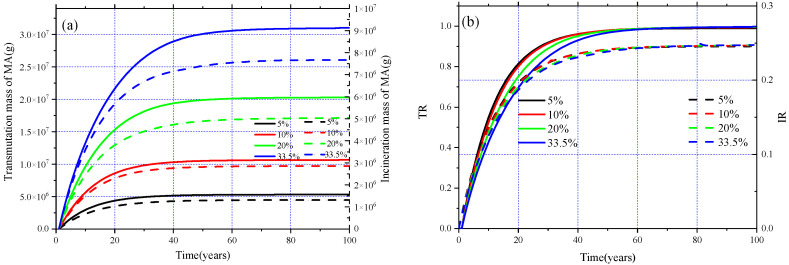
Evolution of transmuted and incinerated MA (**a**), TR, and IR (**b**) for different MA loadings.

**Figure 18 materials-15-08555-f018:**
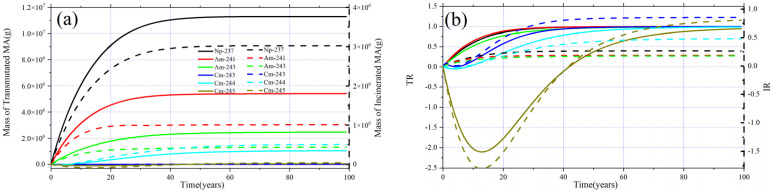
Evolution of transmuted and incinerated mass of single MA (**a**) and TR and IR (**b**) of MA when MA = 10%.

**Figure 19 materials-15-08555-f019:**
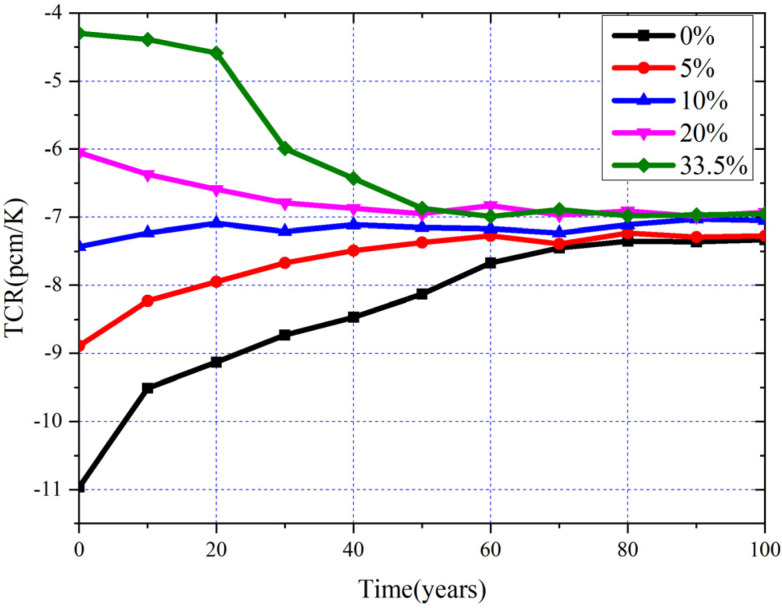
Evolution of TCR for different MA loadings.

**Figure 20 materials-15-08555-f020:**
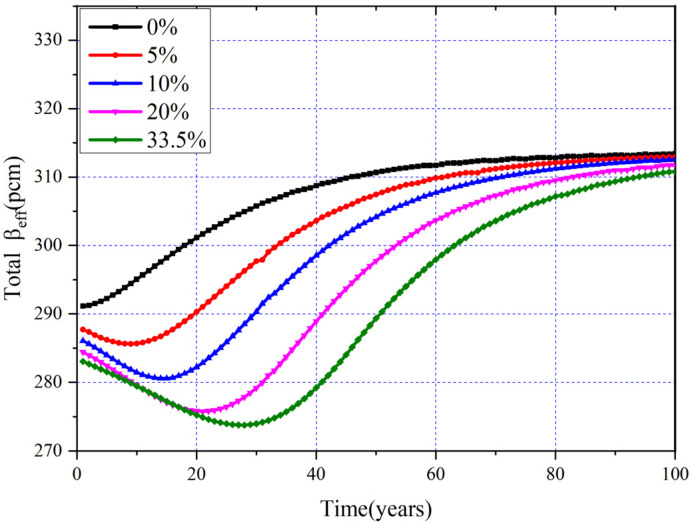
Evolution of *β_eff_* and *β*_s_ for MA = 5%.

**Figure 21 materials-15-08555-f021:**
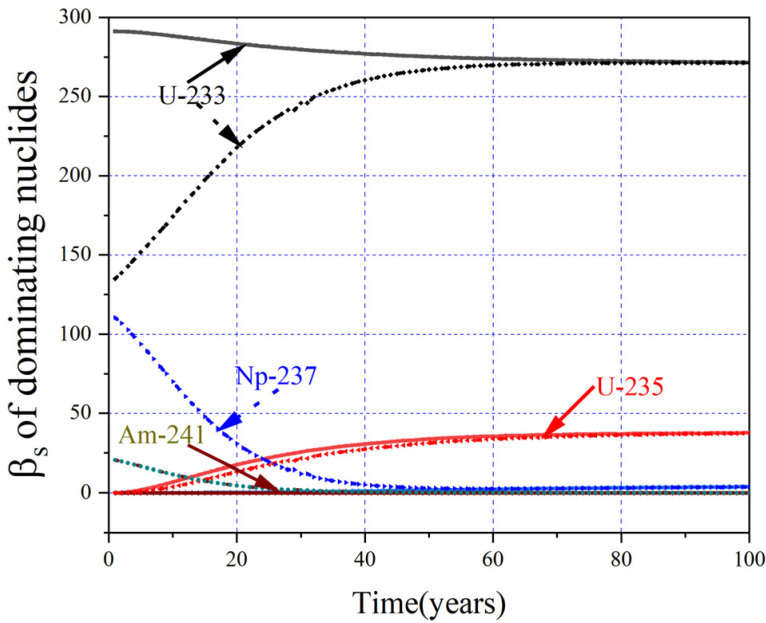
*β_s_* of main nuclides for MA = 0% (solid lines) and MA = 5% (dash lines).

**Table 1 materials-15-08555-t001:** Parameters of the optimized MCFR.

Parameters	MCFR
Thermal power density (MWth/m^3^)	100
Composition of fuel salt (mol%)	NaCl:(HM)Cl_4_ = 55:45
Enrichment of ^37^Cl (%)	97
Temperature of fuel salt (K)	923
Density of molten salt (g·cm^3^)	3.60
Density of alloy (g·cm^3^)	8.86
Density of B_4_C (g·cm^3^)	2.52
Volume (m^3^)	25
Thickness of graphite (cm)	40
Thickness of B_4_C (cm)	35
Thickness of blanket (cm)	60
Thermal expansion (K^−1^)	−3.00 × 10^−4^

**Table 2 materials-15-08555-t002:** Actinide weight ratios of MA [2,37].

Nuclides	^237^Np	^241^Am	^243^Am	^243^Cm	^244^Cm	^245^Cm
Ratio (%)	56.20	26.40	12.00	0.03	5.11	0.26

**Table 3 materials-15-08555-t003:** Transmutation performance of MA with different MA loadings.

MA Loading (%)	MA Transmutation Ratio (%) MA Incineration Ratio (%)						*k* _eff(BOL)_	Life (Years)
	^237^Np	^241^Am	^243^Am	^243^Cm	^244^Cm	^245^Cm		
0	-	-	-	-	-	-	1.208	8
	-	-	-	-	-	-		
3	46.72	50.75	40.14	−5.23	−4.87	−186.77	1.158	10
	12.48	13.94	10.91	−4.57	−2.89	−161.91		
5	77.02	80.79	68.47	40.91	21.08	−218.54	1.136	19
	20.61	22.19	18.61	35.68	12.51	−189.44		
7.5	90.98	92.94	83.77	78.43	53.62	−133.07	1.114	34
	24.34	25.54	22.76	68.39	31.82	−115.37		
10	93.28	94.73	86.69	84.43	61.86	−98.23	1.097	38
	24.93	26.02	23.55	73.63	36.71	−85.17		
15	94.09	95.36	87.84	86.73	65.71	−76.81	1.082	42
	25.18	26.20	23.85	75.63	39.01	−66.59		
20	95.06	96.54	89.21	89.03	69.89	−55.01	1.085	47
	25.44	26.43	24.23	77.63	41.47	−44.73		

**Table 4 materials-15-08555-t004:** Required ^233^U loading at the critical condition.

MA Loadings (%)	Mass of Nuclides (kg)			^233^U Loadings (%)
	^233^U	^232^Th	MA	
0	5.32 × 10^3^	4.14 × 10^4^	----	5.11
1	5.34 × 10^3^	4.03 × 10^4^	1.06 × 10^3^	5.14
3	5.27 × 10^3^	3.84 × 10^4^	3.19 × 10^3^	5.06
4	5.27 × 10^3^	3.73 × 10^4^	4.24 × 10^3^	5.05
5	5.21 × 10^3^	3.61 × 10^4^	5.31 × 10^3^	4.99
7	4.93 × 10^3^	3.43 × 10^4^	7.44 × 10^3^	4.73
10	4.69 × 10^3^	3.14 × 10^4^	1.06 × 10^4^	4.54
20	3.22 × 10^3^	22.3 × 10^4^	2.12 × 10^4^	3.10
30	1.16 × 10^3^	1.44 × 10^4^	3.19 × 10^4^	1.12
33.5	----	1.11 × 10^4^	3.56 × 10^4^	0.00

**Table 5 materials-15-08555-t005:** The *β_eff_* for a single nuclide (*β_I_*).

Nuclides	*β_I_* (pcm)				
	0	5	10	20	33.5
^233^U	291	291	290	290	288
^237^Np	400	398	396	396	394
^241^Am	136	135	135	134	134
^243^Am	239	237	237	237	236
^243^Cm	86	86	86	86	85
^244^Cm	131	130	130	130	129
^245^Cm	177	176	175	175	175
^235^U	669	668	668	667	665

**Table 6 materials-15-08555-t006:** *β_eff_* for different MA loadings.

MA Loadings	*β_eff_* (pcm)				
	BOL	Minimum	Maximum	Fluctuation	EQL
0%	291	291	313	22	313
5%	288	285	313	28	313
10%	286	281	313	32	313
20%	284	275	312	37	312
33.5%	282	273	310	37	310

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
