# Peer review of "Minor Actinides Transmutation Performance in a Closed Th–U Cycle Based on Molten Chloride Salt Fast Reactor"

_materials, 2022, doi:10.3390/ma15238555_

Round 1
Reviewer 1 Report
1 The objective of the study should be clearly given.
2 Remove “on” from the title.
3Recast lines 27-29 and 203-204.
4Remove lines 95-97.
5Line 152: Main should be replaced by mainly.
6The outcome of this study is not forthcoming.
7Author’s distinct contribution resulting from this study should be properly highlighted.
8 Implications of results should be clearly amplified to enhance value of this work.
9On the whole, the manuscript needs a very serious editing for improvement of language, coherence, clarity and presentation also to eliminate repetition.
Author Response
Reviwer1:
Thank you for your careful reading and valuable suggestions, the suggestions are very professional and gave me a lot of new ideas. The following are my answers to your questions and suggestions.
Q1: The objective of the study should be clearly given.
A1: Thank you for your advice, the objective of this study is clearly given in the last paragraph of the introduction, and I highlight it.
Q2: Remove “on” from the title.
A2: Thank you for your advice, I have deleted “on” from title.
Q3: Recast lines 27-29 and 203-204.
A3: Thank you for your suggestion, I have recast lines 27-29 and 203-204.
Q4: Remove lines 95-97.
A4: Thank you for your advice, I have removed lines 95-97.
Q5: Line 152: Main should be replaced by mainly.
A5: Thank you for your advice, it is my error, I have checked it and revised.
Q6: The outcome of this study is not forthcoming.
A6: Thank you for your valuable advice. I guess what you mean here is that the MCFR involved in the paper has not been developed to the commercial stage at present, so the research outcome is not forthcoming. If my understanding is wrong, please do not hesitate to tell me, thank you.
First of all, I would like to express that the MCFR is indeed in the stage of pre-concept design. However, MCFR is also gaining more and more favor due to its unique characteristics. At present, France, the United Kingdom, Germany, the United States, Russia and other countries are conducting in-depth research on it. For example, Terra Power, the company of Bill Gates, a winner of the Advanced Reactor Demonstration Program risk-reduction pathway with Molten Chloride Reactor Experiment (MCRE) proposal selected by the U.S. Department of Energy. It now only working on an Integrated Effects Test (IET) to learn how the MCFR technology will scale and behave at larger, commercially relevant sizes. The IET is expected to be commissioned and begin operating in TerraPower’s Everett, Washington, facility in 2022 [1]. Thus, I think MCFR has its own unique advantages and application prospects as well as the possibility of being built in the future. In the previous, we completed such work as Cl-37 enrichment analysis, evaluation and selection of various pre-concept design schemes, and multi-objective optimization for its breeding capability. Here we focus ourselves on evaluating its potential of transmuting MA and breeding 233U simultaneously on different operation modes, and explore a new Th-U cycle mode in addition to using 233U, 235U or 239Pu as starting fuel [2].
Q7: Author’s distinct contribution resulting from this study should be properly highlighted.
A7: Thank you for your suggestion, I have given the distinct contribution from this research in the conclusion and highlighted it.
Q8: Implications of results should be clearly amplified to enhance value of this work.
A8: Thank you for your advice, in the conclusion, in addition to the analysis of research results based on the optimized MCFR, I summarize some general conclusions which can be generalized to other reactors.
Q9: On the whole, the manuscript needs a very serious editing for improvement of language, coherence, clarity and presentation also to eliminate repetition
A9: Thank you for your careful reading and evaluation. I have reviewed and revised the manuscript comprehensively, and the article will also be polished by professionals of the journal.
References
- Molten Chloride Fast Reactor Technology - TerraPower
- D. Y. CUI et al., “Possible Scenarios for the Transition to Thorium Fuel Cycle in Molten Salt Reactor by Using Enriched Uranium,” Prog. Nucl. Energy, 104, 75 (2017);
https://doi.org/10.1016/j.pnucene.2017.09.003.
Reviewer 2 Report
Minor actinide transmutation is a popular and indeed promising approach to reducing the long-term radioactivity of generated radioactive waste and could lead to the rejection of underground waste storage facilities, including justifying their safety for many thousands of years. The relevance of the thorium-uranium fuel cycle is also unquestionable, given the thorium reserves, which significantly exceed the uranium reserves for the uranium-plutonium cycle, as well as the reduction of long-lived nuclides in the spent fuel. For these reasons, the article is interesting, providing a rationale for involving MA in this cycle. I believe that the article can be accepted for publication after minor revisions.
Comments:
It is appropriate to focus in the Introduction section on the intended spent fuel salt reprocessing technology, and to compare reprocessing technologies for the MCFR reactor and the molten fluoride reactor (e.g., full discharge of spent salt, bypass online salt purification, purification conditions, etc.). Ultimately, it is the salt processing technology that could be the cornerstone in the large-scale implementation of molten salt reactors. It also makes sense to indicate whether the materials chosen to design the MCFR reactors given the aggressive operating conditions.
Most of the figures are at too small a scale and have poorly legible symbols, scale captions, and curves (e.g., Figures 14-18). It is recommended that the figures be converted to a reader-friendly format.
Author Response
Reply to Reviwer2:
Thank you for your careful reading and valuable suggestions, the suggestions are very professional and benefited me a lot. The following are my answers to your suggestions.
Q1: It is appropriate to focus in the Introduction section on the intended spent fuel salt reprocessing technology, and to compare reprocessing technologies for the MCFR reactor and the molten fluoride reactor (e.g., full discharge of spent salt, bypass online salt purification, purification conditions, etc.). Ultimately, it is the salt processing technology that could be the cornerstone in the large-scale implementation of molten salt reactors. It also makes sense to indicate whether the materials chosen to design the MCFR reactors given the aggressive operating conditions.
A1: Thank you for your suggestion. I quite agree with your opinion, reprocessing technology is the key to the popularization of MSR and the realization of closed fuel cycle, so I revised introduction and compared the differences in reprocessing methods between the MCFR and molten fluoride salt reactor, and highlighted it using the yellow color.
Q2: Most of the figures are at too small a scale and have poorly legible symbols, scale captions, and curves (e.g., Figures 14-18). It is recommended that the figures be converted to a reader-friendly format.
A2: Thank you for your advice. I'm really sorry that the clarity of some pictures was reduced when they were converted to PDF. Therefore, I redrew most of the images and marked them with yellow in the article
Round 2
Reviewer 1 Report
The paper is recommended for publication.